# Disseminating STEM Subjects and Ocean Literacy through a Bioinspired Toolkit

**DOI:** 10.3390/biomimetics8020161

**Published:** 2023-04-17

**Authors:** Daniele Costa, Laura Screpanti, David Scaradozzi

**Affiliations:** 1DIISM—Department of Industrial Engineering and Mathematical Sciences, Università Politecnica delle Marche, 60131 Ancona, Italy; 2DII—Department of Information Engineering, Università Politecnica delle Marche, 60131 Ancona, Italy; l.screpanti@univpm.it (L.S.); d.scaradozzi@univpm.it (D.S.)

**Keywords:** STEM education, ocean literacy, educational robotics, robotic fish, robotic toolkit

## Abstract

Over the last decade, education has been evolving to equip students with the fundamental skills required to cope with the challenges of sustainability and inclusivity, such as quality education, access to clean water, cultural heritage preservation and protection of marine life. Technology supports the learning process by providing useful tools that enrich the learning environment, encourage active participation, improve collaboration and prepare students for their future life. Educational Robotics is one of the most popular innovative methodologies that supports the development of many skills by assembling and programming robots in a meaningful way. In this paper, the authors aim at advancing their previous work in the field of Educational Robotics applied to the marine environment by proposing a novel bioinspired educational toolkit whose design and features support activities concerning sustainability, ocean literacy, as well as STEM subjects in kindergarten through to grade twelve education. Exploiting the established educational theories and methodologies underpinning Educational Robotics, the toolkit allows for marine-themed activities, as well promoting activities concerning STEM subjects. To explain the relevance of the toolkit, the authors present the robot design, the workshops that every teacher or student can explore as an Open Educational Resource (OERs), and the results of a case study. Interestingly, the latter shows that the use of the toolkit seems to have complemented the students’ initial keen interest in technology itself, with awareness about urgent issues related to the climate and the environment.

## 1. Introduction

Education has been evolving to adapt to the many requests of a fast-changing world. The aim of the educational changes is not for the sake of innovation itself, but to equip students with the fundamental competences they will need to cope with the overarching challenges of sustainability and inclusivity, including clean air and energy, quality education, access to clean water and protection of aquatic life [1].

Technology can play a fundamental role in reaching a sustainable and inclusive future, and nowadays, many technological tools support the development and implementation of educational activities inside and outside the classroom. Educational Robotics (ER) seems to be one of the most popular trends in education since Papert’s pioneering work [2,3].

Many studies report the effectiveness of ER to teach robotics and coding or curricular subjects such as math, physics [4,5] or general STEM education [6]. In addition, ER has been reported to develop computational thinking [7] and skills such as communication, negotiation, teamwork [4,5,6], while supporting an a more equitable education [8] that involves students in an inclusive technological environment from early education to combat gender stereotypes [9]. Notably, ER has been proven to be effective in supporting environmental education in a range of different contexts [10,11,12,13,14,15]. For example, Ziouzios et al. [15] proposed a robot delivering a story about the dramatic effects of climate change to foster environmental empathy among primary school students. Moreover, Arnett et al. [12] proposed an interactive robotic trash bin disguised as a dinosaur to teach how to recycle different materials to primary school students. Scaradozzi et al. [10] designed and implemented an educational path that let fourth grade students explore the theme of increasing the sustainability of their city using technology. Furthermore, Ruiz-Vincente et al. [14] designed and implemented an activity whose aim was to explain sustainability to fifth graders using the concept of a smart city. Talib et al. [13] observed that many efforts in the integration of robotics in secondary education focused on enhancing students’ skills and knowledge. However, using robotics to facilitate learning in science and engineering applications combined with environmental education could help promote a sustainability mindset that is much needed to face the challenges of the future.

Although some ER activities about environmental education have already been implemented in K12 education (from kindergarten to grade twelve, it usually includes kindergarten, primary and secondary education), only a few of them focus on the marine environment.

Usually, marine robotics activities are carried out in higher education programs, where students learn the knowledge and develop the skills they will need in their future work. However, the existing literature about ER applied to the marine environment highlights that it can be integrated also to primary [11,16], middle [17,18,19] and high school education [18,19]. 

Marine Robotics can be useful to attract students to blue careers [19,20,21] and to talk about ocean education with students with special needs [22]; however, it is applied mainly in non-formal education and less in formal K12 education. Several competitions exist to involve students in educational marine robotics [19,21,23,24,25], as well as informal science education activities at the museum [26,27]. However, only [11] proposed an integrated curriculum of activities about ER and marine environment dedicated to primary education which was also suitable for higher grades.

Moreover, the toolkits used in existing non-formal activities about marine robotics and ER are Remotely Operated Vehicles (ROVs) or Autonomous Unmanned Vehicles (AUV) [16,19,21,26,27]; only Phamduy et al. [28] proposed a fish-like robot. On the contrary, authors in [11] proposed and piloted a set of organized activities that could adapt to formal education supported by the common commercial toolkits and a fish-like robot that could be programmed thanks to the available commercial educational toolkits. One of the reasons for the limited number of experiences about marine robotics in education can be traced to the substantial lack of suitable commercial toolkits composed by a set of mechatronic parts that have all the following features: they can actually be assembled, programmed and submerged into the water, are commercially available with a low cost and provide the lesson plans that explain how to build activities using the toolkit [11].

The present paper aims at advancing the work of [11], proposing a novel tool for K12 education to bring together sustainability, ocean literacy and the educational methodologies underpinning ER. The design of the new tool can exploit the wide range of benefits brought by the ER methodology, while carrying out marine-themed activities and learning STEM subjects. To explain the relevance of the tool, authors present not only the design of the robot (Section 2), but also an educational path about educational marine robotics that every teacher or student can explore (Section 3), in addition to a case study (Section 4). The educational path presented in Section 3 was developed to support the teachers of the Erasmus+ Robopisces project (“Innovative Educational Robotics strategies for Primary School Experiences”, 2019-1-IT02-KA201-063073) [29]. The case study presented in Section 4 was carried out thanks to the support of the Fondi Strutturali Europei—Programma Operativo Nazionale “Per la scuola, competenze e ambienti per l’apprendimento” 2014-2020.

The paper is organized in the following sections: Section 2 shows the main features of the toolkit; Section 3 highlights the concepts and activities that can be adapted to K12 and higher education; and Section 4 illustrates a summer camp experience carried out in Italy providing an example of the proposed approach.

## 2. Toolkit Design and Manufacturing

The bioinspired toolkit presented in this paper is shaped like a fish and is capable of propelling itself by a spinning the tail section hinged to rigid forebody which mimics ostraciiform swimmers [30]. Ostraciiform locomotion is the least efficient among body and caudal fin swimming modes. However, since the oscillating tail is the only moving part, the robot architecture is very simple, inexpensive, easy to seal, fabricate and maintain, even by untrained primary school personnel. These features have a higher priority with respect to more efficient locomotion systems. In fact, aside from the capability to move and perform turn maneuvers into a pool or a tank, a robotic toolkit designed to be operated in primary school classes must meet the following requirements:The robot controller must be programmed and operated remotely, meaning without opening the sealed section where the onboard electronics is housed;In the same way, the batteries must be recharged without opening the sealed compartment housing the energy system;The controller buttons and display must be accessed from the outside;In case of failure, the robot servomotors must be replaced by untrained personnel;The robot must meet the high safety and environment protection requirements dictated by the Machine Directive, 2006/42/EC, and the guidelines provided by the Toys Safety Directive (TSD), 2009/48/EC.

On the basis of the stated requirements, although the robot propulsive performance would have severely improved through the adoption of a more complex design such as a multi-joint tail mechanism mimicking carangiform or thunniform swimming, the authors have chosen to focus on the simplest architecture and the resulting ostraciiform locomotion in order to craft a solid and inexpensive robot that can be exploited as a bioinspired toolkit for primary school workshops. As a matter of fact, the aim of the whole project is not to design the most efficient or a high-performance prototype, but to provide a robust, fully functional product which is easy to manufacture in large numbers in order to be distributed to as many public schools as possible.

Figure 1a shows the blueprints of the robotic fish presented in this paper: the complete assembly is 350 mm long, 50 mm wide, and has a mass of 550 g when neutrally buoyant. The features implemented to meet the requirement stated above are described as follows.

### 2.1. Propulsive and Maneuvering Systems

The robot thruster consists of an oscillating tail and a flat caudal fin attached to it. The assembly is driven by a waterproof servomotor embodied in the tail, which is manufactured as a thin hollowed shell. As stated before, and according to the nomenclature commonly adopted in the literature, the robot swims exploiting ostraciiform locomotion. Here, the tail performs harmonic oscillations, while the momentum transfer due to the relative motion between the caudal fin and the surrounding water generates the necessary thrust to balance the drag force. When the tail neutral position coincides with the robot vertical plane of symmetry, the average lateral force generated by the caudal fin in an oscillation cycle is null. On the contrary, when the tail neutral position forms a constant bias angle with the vertical plane, as shown in Figure 1b, the thrust force has a centripetal component which allows the robot to steer.

In order to size the tail servomotor and the caudal fin, the following equation can be written to balance the propulsive and resistance forces:(1)12ρU2AfCD=12ρU2CTSfin
where *ρ* is the water density, *U* is the cruising speed, *A_f_* is the robot frontal area, *C_D_* is the corresponding drag coefficient, whereas S*_fin_* and *C_T_* are the fin surface and average thrust coefficient. In [31], the authors of this paper have shown that the average thrust coefficient of a flat fin spinning according to a harmonic motion law may be expressed as:(2)$$C_{T}=K_{T}\left(\theta_{0}\right)St^{2}$$
where *K_T_* depends on the fin oscillation amplitude *θ*_0_, whereas *St* is the Strouhal number, a dimensionless parameter commonly used in oscillating-flow phenomena:(3)$$St=\frac{2c\sin\theta_{0}}{U}f$$
where *c* and *f* are, respectively, the fin chord length and oscillation frequency. Figure 2 shows the trends of the average thrust coefficient *C_T_* predicted as a function of *St* and *θ*_0_. By replacing the Strouhal number expression (3) in (2), and the latter in (1), the following sizing equation can be written:(4)$$U^{2}=f^{2}\frac{S_{fin}}{A_{f}}\frac{K_{T}\left(\theta_{0}\right)}{C_{D}}4c^{2}\sin\theta_{0}{}^{2}$$
where the optimal value of *θ*_0_ has been figured out by the authors in [32]. Then, according to the blueprints of Figure 1a, the average cruising speed U can be computed by means of expression (4). Table 1 summarizes the resulting geometric and kinematic parameters.

The chosen actuator is a Traxxas 2065A waterproof digital micro servo manufactured by Power-HD, which is capable of spinning at 1.67 Hz and supplying a stall torque of 0.26 Nm.

Although the robot architecture is simple, its swimming performance has been deeply investigated through a dynamic analysis performed using the design and simulation platform developed by the authors in [33]. To this end, the robot CAD model has been imported in the multibody environment provided by MSC Adams View, as shown in Figure 3. Here, the model inertia is the same as the neutrally buoyant robot. Regarding the hydrodynamic loads, both added mass and damping forces have been applied to the model body and tail. Finally, the propulsive loads generated by the tail motion have been applied to the caudal fin using run-time functions to compute their modulus through the simulations. Further details of the dynamic model and loads expressions are provided in Appendix A. Since the aim of this analysis is to investigate the robot swimming performance both in cruising and maneuvering modes, a planar joint has been applied to the model center of mass to constrain its motion to the horizontal plane.

The first set of simulations relates to the cruising performance: here, the servomotor frequency has been varied in a suitable range, whereas the tail oscillation amplitude has been left constant at the optimal value. Table 2 collects the results, showing the average steady cruising velocity achieved at the end of the acceleration transient, together with the maximum torque required to spin the tail at different frequencies. By comparing the latter with the torque available from the servo, which has been reported in the rightmost column of Table 2, it can be seen that the robot is capable of swimming relative to the design conditions of Table 1. However, as the tail oscillation is increased, a progressive performance reduction is foreseeable due to the impossibility to provide the necessary torque.

The final set of simulations investigates the robot maneuvering performance. Here, the steering angle, meaning the constant bias angle between the tail neutral position and the robot vertical plane of symmetry, has been increased up to 45 degrees. Table 3 collects the results, showing the turnabout radiuses measured at different values of the cruising speed. The results are expressed in multiples of the robot body length (BL).

### 2.2. Safety Features and Manufacturing

The forebody section comprises two parts, as shown in Figure 1a: the head, a 2 mm thick hollowed shell shaped to resemble a carp, and the body, a waterproof compartment housing the robot controller and its energy systems. The watertight connection between the body and its elliptical cap is sealed by a nitrile rubber (NBR) O-ring housed inside the radial groove carved inside the cap. Four self-locking bolts prevent the students, as well as other untrained personnel, from accidentally opening the body or tampering with the sealing. The waterproof compartment has been successfully tested in a pressure chamber up to a 10 m depth. Regarding the robot head, the hollow shape has been chosen in order to allow the students to practice with buoyancy and aptitude, as will be detailed in Section 3. In fact, the head part, as well as the tail described in the previous subsection, are meant to be flooded when the robot is immersed; hence, the only positive component is the sealed body, which produces enough buoyancy to prevent the robot from sinking. The head internal surface can be easily accessed by removing the four screws connecting to the body. Once disassembled, known weights can be placed in the slots created on the internal surface, thus varying the toolkit mass and mass center position.

As stated before, the robot controller must be programmed and fully operated when the body is closed. This feature has been chosen to prevent the students from opening the toolkit, which could result in water leakages in case of incorrect assembly. As a matter of fact, the toolkit has been designed to remain closed straight from the box, unless maintenance is required due to components failures. The controller, a M5StickC supplied by M5Stack, can thus be programmed by establishing a Wi-Fi connection with a computer or smartphone running the development environment application described in Section 3. Moreover, the controller buttons have been made remote by attaching reed switches to the internal surface of the body; in this way, students can easily switch the buttons by moving a magnet close to the robot’s back, in correspondence of the printed labels. Finally, the controller display can be observed through the porthole sealed on the fish back.

The robot controller and the servomotor driving the fish tail are powered by a nickel–metal hydride battery pack sealed in the body. Although lithium-ion batteries deliver more power compared to nickel–metal hydride, the latter are not as dangerous as the former. As a matter of fact, nickel–metal hydride does not cause a fire when exposed to oxygen. On the contrary, lithium-ion batteries explode when exposed to oxygen. Moreover, when exposed to water, nickel–metal hydride batteries discharge quickly, causing no harm to people or systems. Therefore, they represent the optimal choice for a toolkit dedicated to primary classes. Finally, in order to recharge the batteries when the body is closed, a wireless charging receiver module has been merged into the robot belly, together with its dedicated circuit board which is connected to the batteries. In this way, the robot can be easily recharged by placing it over any commercial wireless battery charger.

Regarding the manufacturing process, the toolkit structural components have been 3D printed by means of high precision stereolithography. In order to meet the high safety requirements stated before, the authors employed a resin which has been evaluated as a skin contacting device in accordance with ISO 10993-1, and classified as non-cytotoxic, non-irritant, and non-sensitizing in the post-curing state in accordance with ISO 10993-5, ISO 10993-10. Furthermore, the toolkit has been carefully drawn to prevent the possibility of finger trapping between the moving parts. Particularly, the maximum clearance between the body and the tail is 1.5 mm, noticeably lower than the 4 mm threshold dictated by the TSD. Figure 4 summarizes the assembly and the safety features described in this Section.

Finally, the toolkit underwent a risk assessment which was carried out by a licensed company in accordance with the current standards. It was evaluated to comply with a high standard of safety and environmental protection requirements, as dictated by the Machine Directory. Therefore, it was worthy of being sold as a product bearing the CE marking.

## 3. Curricula and Workshops

Usually, primary school teachers have little practical knowledge in subjects such as programming and mechatronics. Therefore, to enhance their skills and understanding the Erasmus+ RoboPisces project conceived a comprehensive online course presenting teachers with Open Educational Resources (OERs), such as videos, a forum, a slideshow presentation, and demo software. The teachers could use any mechatronic device to follow the training and to prepare their practical activities. Nevertheless, the bioinspired toolkit presented in the previous section embodies all the features presented in the following lessons; hence, it represented the optimal hardware and software tool for the training course.

The whole course is presented two main parts: the basic topics about the fundamentals of robotics and the advanced topics about IoT (Internet of Things) and the marine environment (Figure 5) [29]. The whole plan of the course was called the “FISH curriculum” and it identified the main topics about technology that can support a two-year curriculum at school. The content of both the basic and advanced topics of the course, in fact, can be adapted to cover one year of theoretical lessons and workshops activities for primary school students. In the first part of the course, the orange and green modules of Figure 5 were covered. The curriculum introduced the fundamental concepts about robots and machines (green topics). At the same time, it facilitated a better understanding of the educational methodology in relation to the broad field of robotics, which provides the context of the activities (orange topics). Providing an overview of robotics and their applications is a fundamental aspect of the educational process, as it can help set up meaningful activities for students. For example, introducing the robot life cycle helps learners reflect on how to design, fabricate, maintain and dispose robots, and why it is important to develop engineering skills that support that process. The first part of the course also provided insight on the various mechatronic components of the bioinspired toolkit, such as sensors, motors and its main controller, including the programming environment where simple tasks and complex missions can be coded. In the second part of the curriculum, which focuses on the environmental aspects (blue topics of Figure 5) and IoT concepts (peach topics of Figure 5), learners built upon the knowledge and skills developed in the previous topics. The bioinspired design of the toolkit is used to implement activities about the marine environment and ocean literacy, including concepts such as the blue economy and cultural heritage conservation. Physics also played an important role throughout the marine robotics activities: static and dynamic equilibrium in the aquatic environment were discussed and analyzed, whereas the bioinspired toolkit helped the teachers become familiar with concepts such as buoyancy, balancing, recoil, thrust and drag. The IoT concepts covered the fundamentals of communication and fleet management for robots, using the example of a school of fish. Teachers of the RoboPisces project explored the whole curriculum at their own pace by accessing the e-learning platform where all the contents were stored (www.robopisces.eu/robopisces-mooc (accessed on 20 February 2023)). The teacher training about the basic course started in August 2020, whereas the training about the advanced course started in July 2021. In terms of time, the effort required by teachers to complete the basic course was estimated to be needing 5 days, whereas the advanced course was estimated as needing 15 days.

### 3.1. Basic Course Lessons and Workshops

The basic course (the orange and green topics in Figure 5) opened with a theoretical lesson which pointed out the similarities and differences between robots and machines: robot autonomy was thoroughly explained and the possibility of programming a robot in order to perform different tasks was compared with the lack of a possibility to exceed the limits imposed by their fabrication, which normally limits them to a single function, thus providing the machine with a defining characteristic. Although both robots and machines are capable to sense the surrounding environment, as well as elaborate signals and information, only robots can autonomously implement suitable reactions. This idea, strengthened by the concepts of inputs and outputs, was the backbone of the whole curriculum, and was extensively exploited throughout the lessons and workshops. Regarding the practical part, the teachers were trained in particular to organize a simple contest where students were asked to provide examples of machines and robots from their daily experience.

The role of the roboticist was the focus of the second lesson. Here, the teachers were introduced to the different engineering skills required to design, fabricate, program, repair and dispose of a robotic system at the end of its life cycle to minimize its impact on the environment. During the practical session, the teachers engaged in a roleplaying game that they could potentially introduce to primary school students. The purpose of the game is to show how the engineering skills mutually interact when a new robot is designed. Individual tasks were thus assigned to the players, which in turn, were asked to make decisions regarding the design, manufacturing and deployment of the robotic system.

In the third class, the teachers were involved in a three-module session represented by the green tags of the FISH curriculum and that was dedicated to the functional blocks already introduced in the first lesson—sensors, actuators, and the controller, which began with the latter. A short presentation was dedicated to the controller which was chosen for the bioinspired toolkit shown in the previous section—the M5StickC produced by M5Stack. Nevertheless, the core of this lesson was the hands-on activity where the teachers were trained on the fundamentals of coding. Supported by the Integrated Development Environment (IDE) based on Blockly, an open-source software released by Google, the teachers learned how to create projects by means of a visual programming editor running on Windows and Android web browsers, which allows blocks of code to be dragged and dropped from toolboxes and palettes and arranged in the workspace. Additional blocks of code specific to the hardware devices supplied by M5Stack were also provided. Several tasks and exercises were assigned to the teachers in the practical session to practice with the most common features of coding, such as loops, input and output implementation, buttons, turning on LEDs, and so on.

The second lesson of the three-module session was dedicated to sensors. Following an introduction about exteroceptive and proprioceptive sensors, the teachers were asked to solve quizzes about self- and environment awareness. The class also highlighted the analogies between robotic detection and human perceptions, providing several examples of commercial sensory devices such as cameras, probes and microphones which mimic the behavior of human vision, hearing, smell, taste and touch. In the following workshop, the teachers were asked to exploit the knowledge gained in the preceding lesson to implement a code which could measure the attitude of the M5StickC by means of its Inertial Measurement Unit (IMU). The assigned task also included the printing of an arrow, working as a bubble level, on the controller display and the computation of the vertical acceleration.

The final lesson of the three-module session, as well as the basic curriculum, focused on actuators. In this class, the teachers were introduced to the analogy between the human musculoskeletal system and the electrical or pneumatic actuators commonly used in robotic systems. The teachers also became familiar with the fundamental concept of motor drives: just as muscles contracts when a bio-electrical signal is passed from the brain through the nervous system, in robots, motors spin when reference signals are sent from the robot controller and then converted in driving signals, such as Pulse-Width Modulation (PWM), by the motor drive. The theoretical class also focused on the purpose of actuators, such as manipulation and propulsion. In fact, motors drive robotic arms, tools, wheels and thrusters similarly to how muscles power arms, legs, hands and feet. Later, in the following workshop, teachers were asked to connect the M5StickC board with the servo drive produced by M5Stack, and then develop proper code to generate servo oscillations.

### 3.2. Advanced Course Lessons and Workshops

The advanced course opened with the most important lesson of the FISH curriculum—ocean literacy, which is the understanding of our individual and collective impact on the ocean and its impact on our lives and wellbeing. The theoretical session focused on the importance of aquatic environment preservation and on the role of water for life sustainability on our planet, which is definitely one of the critical issues of modern society. The teachers were also introduced to the seven principles of ocean literacy, where they learned how weather and climate are influenced by the ocean, how life and ecosystems are supported by the aquatic environment, how the blue economy works and how human beings are obviously inextricably interconnected to the ocean. During this session, the teachers were taught about the importance of preserving cultural heritage, with a focus on how the ocean holds valuable treasures and mysteries that are concealed beneath the waves and buried in archaeological sites, waiting to be unearthed. Later, in the practical session, the teachers were introduced to the bioinspired toolkit and were familiarized with its safety features and maintenance operations, before starting practicing with its controller and tail actuator. These preliminary tasks were performed on a workbench and were preparatory for the upcoming “wet” operations.

The second advanced class was the first of the physics-dedicated sessions. Particularly, the theoretical lesson focused on fluid statics and on the concept of static equilibrium in the aquatic environment. Here, the teachers were introduced to Archimedes’ principle and the conditions under which bodies rest in a stable equilibrium when immersed into a fluid. In the following workshop, the attendees of the course were trained to organize a simple experiment that was devised to allow primary students to practice with buoyancy and aptitude. When the bioinspired toolkit is immersed right out of the box, it floats with the head rising over the water level. Therefore, the assigned task consisted in attaching known weights to the hull until the robot floated leaving only the dorsal fin out of the water and the bubble level displayed on the controller screen indicated that the fish was leveled at a zero-degrees pitch aptitude or, equivalently, aligned to the horizontal plane.

The second physics-dedicated lesson further investigated fluid mechanics by introducing the concept of dynamic equilibrium. Particularly, the teachers learned how thrust is required in order to balance or overcome the drag force exerted by the fluid. The Principle of Action and Reaction was also mentioned in this class. Regarding the practical activity, the teachers were asked to run the code developed in the basic course to drive the servo embodied in the fish tail and push the robot along a planned path, which was straight first and then followed a circle while navigating through a series of buoys floating inside a pool. This activity also allowed learners to experience the Principle of Action and Reaction discussed in the theoretical lesson, by observing the recoil affecting the fish head while swimming.

The last of the physics-dedicated lessons was a workshop, where the teachers learned how to organize a simple speed race. The aim of the contest was to program the toolkit in order to follow a straight path and cross the finish line, which was indicated by a couple of buoys, in the shortest possible time. The main difficulty was to choose the fin rotation amplitude and oscillation frequency, which severely affect the cruising performance. The attendees of the course were welcome to practice by varying both parameters in the tail code, in order to figure out the best combination.

Lessons about IoT concepts build on the lessons about communication that are delivered during the basic course. The first lesson quickly introduced the concept of a smart thing and how it can be connected to other smart things. Many examples were provided to highlight how smart things are already deployed in the world, and how they are extremely useful. Contextualizing the activity makes it meaningful and also aims at raising curiosity for the technological devices that support many aspects of modern human lives. Then, the lesson ended with a hands-on activity about the protocol for the communication among smart devices. The second lesson about IoT explained how to send a message between more than two smart things. A series of five hands-on activities were developed, each with increasing levels of difficulty that gradually guide the learner towards the goal of creating a sensor network. The real-world scenario for the last two activities was programming the behavior of a smart traffic light system at a crossroad and organizing a network of smart buoys that will send information about the surrounding water to the fish-shaped robot. Finally, the last lesson about IoT concerns the exploration of the basics of Control System Theory, in particular concepts of dynamical systems and open loop control and closed loop control. The last hands-on activity aimed at exploring the differences between a centralized alarm system and a distributed one.

## 4. Case Study

The concepts and activities reported in Section 3 can be supported by the tool presented in Section 2. They are conceived as flexible modules for ER activities, so that educators and teachers can design their own educational pathway starting from the conceptual sequence of topics depicted in Figure 5. Each module can either fit multiple lessons to create a robotics curriculum at school, or it can be combined with other modules to design a short educational pathway about ER, the marine environment and IoT.

To test the validity of the tool and activities, in August 2021, authors carried out a pilot project at Istituto Comprensivo Giannuario Solari of Loreto (Italy), Figure 6. Thanks to the collaboration of the school’s staff, 25 students (age range: 10–13, mean age: 11.8 years old; 52% female, 48% male) benefited from the availability of the school’s facilities during the summer. They took part in non-formal activities about robotics and sustainability for 5 h a day, over a 6-day period. Three educators and one teacher led the activities during the summer camp; all of whom were previously trained on the topics of the FISH curriculum, as reported in Section 3. Table 4 highlights the educational pathway which was designed for the summer experience. Each day, students worked on more than one topic of the FISH curriculum (Figure 5) using either the “basic kit” (the main parts of the fish robot, such as the brain/microcontroller, the actuators, the sensors, cables, and programming environment) or the fish robot. In particular, during the first 3 days, students explored the fundamentals of robotics using the basic kit, whereas during the last 3 days, students explored the fundamentals of marine robotics and environmental education, thanks to the use of the advanced fish robot. Each day students participated in a series of hands-on, playful activities, whose initial phase was the presentation of a practical issue to solve with technology, or an example of robot. Thus, each day, they were introduced to the activity by presenting real world applications or issues connected with the tools and the abilities that they were about to explore. For example, during the final 3 days, students learned about the seven principles of the ocean literacy (oceanliteracy.unesco.org/principles) and reflected on how technology could help humans and the sea. Notably, the outcome of this activity was a story about environmental sustainability using the fish robot and other materials. On the final day of the summer camp, the students showcased their story to the school staff and the local community. They shared what they had learned and the projects they had built using the robotics toolkit. The aim of the storytelling activity was to organize the concepts about robotics and sustainability they learned in order to consolidate knowledge and skills, following the theory of social constructivism [34].

Before starting the activities (Baseline, BL) and after the activities ended (posttest, PT) students answered a short questionnaire about student’s interest and self-efficacy [35] in robotics, and engagement in sustainability themes. The aim of the assessment was to find out whether students enjoyed the activity and whether the proposed educational activities combined with the new educational marine robot increased the attention of the students about environmental issues and sustainability. Table 5 reports the items of the questionnaire and the dimension explored by each item.

Answers to the questionnaire showed that most of the students (59%) participated in the summer camp activities because they liked robotics or because they expected to improve their ability with robotics (25%). Robotics activities were already perceived as fun and engaging before the camp started, and the same share of students agreed or strongly agreed to rate robotics activities as fun and engaging (88%) after the end of the experience, as shown in Figure 7. Even if students reported great interest in robotics activities before the summer camp, only 40% of them thought that it would be easy to use the robotic toolkit. After the activities, the share of those students increased to 68%, as shown in Figure 8.

Students participating in the summer camp were interested in technology and robotics and they had previous experience with different kinds of media and devices. This was evident from the answers; for example, some free-text responses cited “improve videos on my YouTube channel”, “invent app and make videos”, and “use a 3D printer”. After taking part in the activities at the summer camp, students seemed to shift their attention from technology itself to what they can do with technology. Some of the sustainability issues that caught students’ attention were: “collect litter”, “observe plants and fishes underwater”, “remove plastics from the ocean”, and “save people during floods”. Figure 9 represents the occurring themes. Notably, at BL, 80% of the students focused their attention on technological tools, whereas at PT, 60% of the students mentioned the marine environment in their free-text response and 28% imagined carrying out their personal project using the fish robot.

The fish robot seemed to engage students’ curiosity: in fact, when asked about the thing that they liked the most about the camp, they reported they liked working with the toolkit (51%), programming (35%) and also working with a different methodology if compared to the one they are used to at school (14%), as shown in Figure 10.

## 5. Conclusions

The educational approach proposed in the paper aims at fostering STEM education, digital skills, and ocean literacy by exploiting smart pedagogies in a constructionist learning environment using a bioinspired robot. This paper introduces the marine environment thematic during ER activities thanks to the bioinspired toolkit, thus presenting students with vital ideas related to sustainability. Notably, the use of the bioinspired toolkit supported student interest, engagement and motivation in participating in the educational activities of the summer camp. Results from the case study showed enthusiasm was raised by the fish-like robot, even among students with a keen interest and previous experience with media and digital devices. While an excessive use of technology may cause students to concentrate more on the tool than the lesson content (known as "digital distraction") or cause oversaturation of interest in the ER activity [4,36,37], the implementation of the fish-like robot successfully diverted student attention away from the robotics and technology, towards the urgent environmental issues at hand. Remarkably, results from the case study demonstrate not only the engagement in environmental issues that the tool promotes, but also the promotion of interest and self-efficacy in robotics. Therefore, the tool successfully maintains the benefits of introducing robotics from K12 education [5,7,9]. This result is very important because those factors help reduce and prevent the risk of school failure and early school leaving (ESL) [38]. In fact, ER is useful for working with students who have expressed an interest in technology, so that the educators can promote students’ comprehension of subjects by using learning agents, and at the same time, the development of cooperation, peer learning and self-efficacy [38].

The fish-like robot helped achieve the target of sustainability. This can relate to the fact that the natural user interfaces are deemed as the most appropriate for engaging students in serious games [39]. Furthermore, animal-like educational robots are recommended for inclusive education since they seem to raise interest across different groups [40].

Future activities envisage the use of the bioinspired tool in K12 curriculum to boost STEM learning, digital skills development, and ocean literacy. To this end, the authors founded the academic spin-off named ANcybernetics [41] in order to promote the toolkit distribution in schools and to train teachers as well. Secondly, economic sustainability was investigated, and the results are presented in Table 6, showing how the manufacturing cost per robot unit was computed.

About 350 mL of resin are necessary to print the fish structure in a 32 h long process. Here, the skin-proofed resin is worth EUR 180 per liter, so the overall structure cost is EUR 62. The total cost of the mechatronic components—meaning the sealed servo and its driver, the robot controller, the batteries, the wireless charger and other minor electronic components—is about EUR 70 overall. Similarly, mechanical components and consumables—such as O-rings, screws, spacers, and the sealing glue—are together worth less than EUR 10. Hence, the manufacturing cost per robot unit is about EUR 142.

The authors estimated that a skilled technician (EUR 60 per hour of work) can assemble three robots per hour, thus raising the manufacturing cost to EUR 162.

Different scenarios were also analyzed for the fish structure manufacturing: for instance, a third-party 3D printing services company estimated about EUR 30 per robot structure. This offer could further decrease when its parts are manufactured by injection molding by a professional facility. However, the initial investment required to produce the molding equipment is worth thousands of euros, which in turn, means that hundreds or possibly thousands of robots need to be sold to raise a return on the investment.

The high cost of the molding equipment is mainly due to the complex shape of the fish body, which requires a three parts-assembled mold. The process could be optimized by redesigning the main components of the robot for injection-molding manufacturing. In fact, Guizzo was designed using the principles of Design for Additive Manufacturing (DfAM) specifically for 3D-printing technology.

Regarding the time and costs of training the teachers to operate the robot and carry on the activities presented in Section 3, the authors estimated that a 1-week (5 days, 5 hours per day) course could be sufficient to prepare unexperienced teachers both on the basics and advanced activities. This training course, conceived for a 25 teacher-class, is worth EUR 50 per hour, meaning EUR 50 per teacher.

In summary, the total current cost of a toolkit and a training course for an unexperienced teacher is EUR 212, or EUR 180 if the robot structure is manufactured by a third-party 3D-printing company (shipping costs not included).

On the other hand, the toolkit final cost is affected by several other factors such as the expenses normally associated to business activity—for example, rental of premises, power consumption, insurance and accountant fees, etc. Secondly, the consultancy cost that is necessary to comply with the Machine and Toy Directives was worth about EUR 4000. Other investments were necessary to equip the authors’ lab with a professional 3D printer, a soldering workstation, a finishing kit and other equipment, which amounted to a total cost of EUR 10,000.

Through calculation of these factors, a rough assessment of the market cost for the toolkit plus the training package should be about EUR 450. This value was computed by considering a production rate equal to three toolkits per week, or 150 toolkits per year, which would coincide with the breakeven point for the ANcybernetics spin-off. With these costs, ANcybernetics already sold five courses, with 25 teachers each, in 2023.

Further adjustments will be evaluated on a year-by-year basis depending on the market response.

## Figures and Tables

**Figure 1 biomimetics-08-00161-f001:**
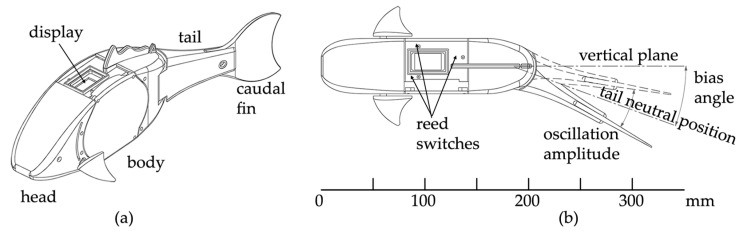
(**a**) Robotic fish blueprints and main components; (**b**) steering configuration.

**Figure 2 biomimetics-08-00161-f002:**
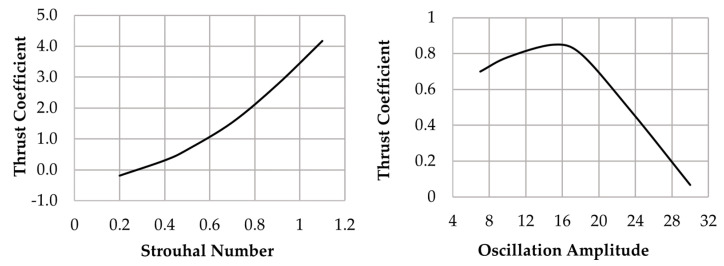
Average thrust coefficient as a function of the Strouhal number and oscillation amplitude.

**Figure 3 biomimetics-08-00161-f003:**
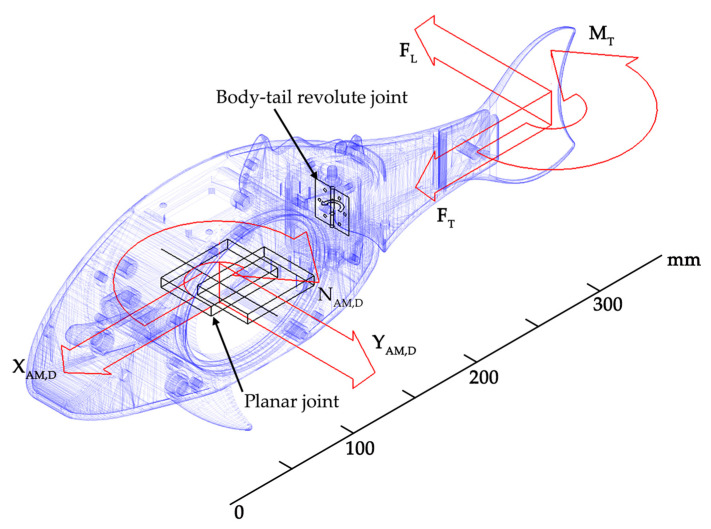
Robot multibody model in MSC Adams View: the hydrodynamic loads are decomposed in the surge (x), sway (y) and heave (z) directions according to the marine vehicle convention. The AM and D subscripts refer to the added mass and damping forces and torque acting on the robot body. The F_T_, F_L_ and M_T_ loads indicate the propulsive forces and torque generated by the caudal fin. The latter expressions are detailed in the Appendix A.

**Figure 4 biomimetics-08-00161-f004:**
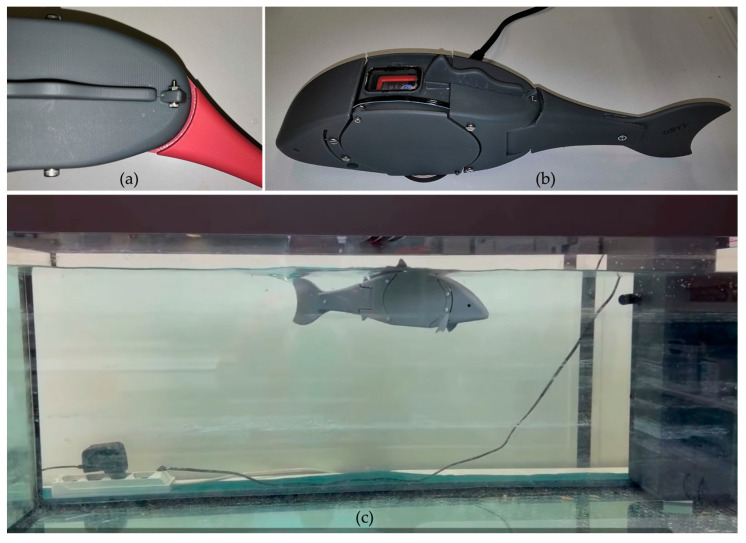
(**a**) Clearance when the tail is rotated; (**b**) batteries charging process, display porthole and active controller; (**c**) robot buoyancy and swimming test in a fish tank.

**Figure 5 biomimetics-08-00161-f005:**
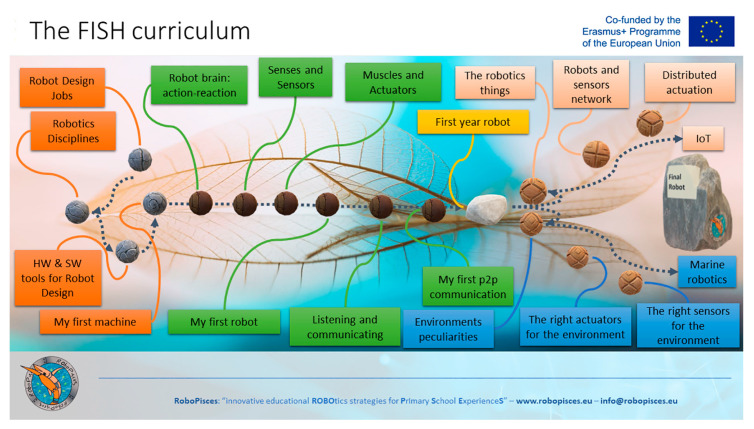
The FISH Curriculum is an educational pathway where each colored label represents a key topic (a module). The basic course is represented by the orange (introductory topics) and green topics (core robotics concepts). The advanced course is represented by the blue topics (marine robotics) and the peach topics (Internet of Things-related concepts).

**Figure 6 biomimetics-08-00161-f006:**
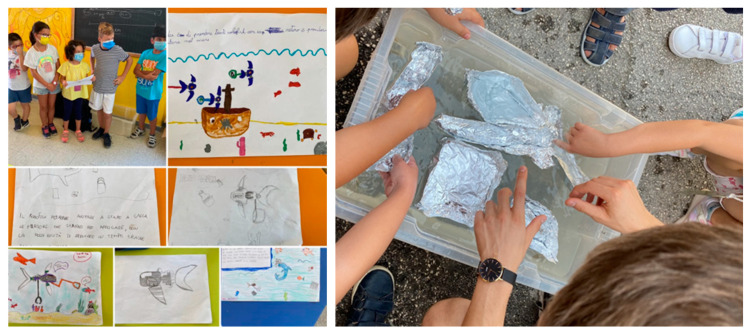
(**Left**): Students rehearsing to represent their story; (**right**): students trying their own surface vessels to understand the concept of “buoyancy”. Tinkering activities were used to introduce useful concepts and also to complement and customize the toolkit itself.

**Figure 7 biomimetics-08-00161-f007:**
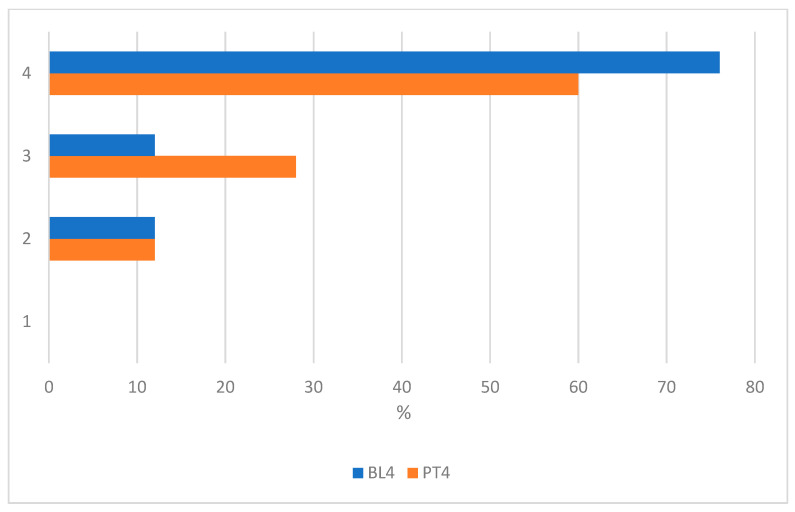
The picture shows the distributions of students’ responses before the start of the activities (Baseline, BL) and at the end of the experience (posttest, PT) about their interest in the activities proposed by the camp. The scale is: 1 (disagree), 2 (somehow agree), 3 (agree) and 4 (strongly agree). The total number of students is n = 25.

**Figure 8 biomimetics-08-00161-f008:**
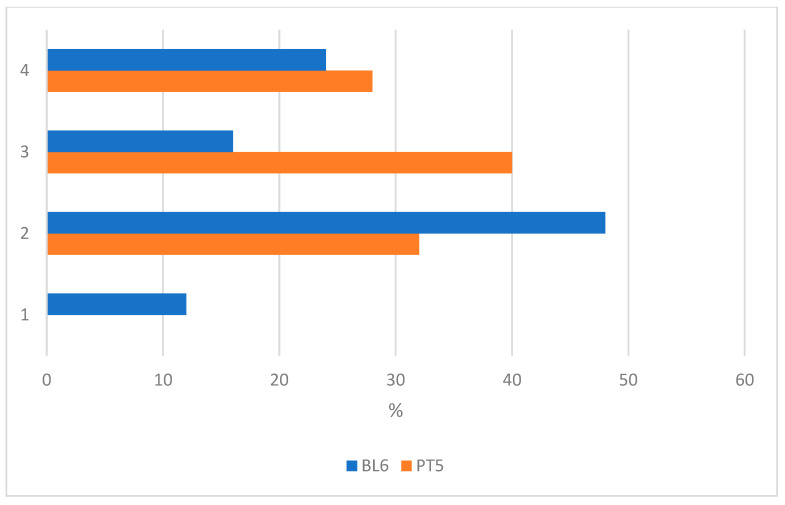
The picture shows the distributions of students’ responses before the start of the activities (Baseline, BL) and at the end of the experience (posttest, PT) about their self-efficacy on robotics. The scale is: 1 (disagree), 2 (somehow agree), 3 (agree) and 4 (strongly agree). The total number of students is n = 25.

**Figure 9 biomimetics-08-00161-f009:**
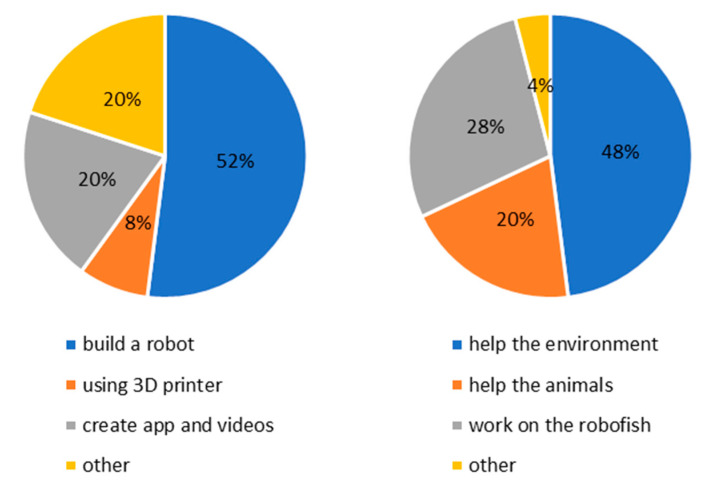
The picture shows the themes of the projects that students would like to carry out before (BL) and after (PT) the activities carried out at the summer camp.

**Figure 10 biomimetics-08-00161-f010:**
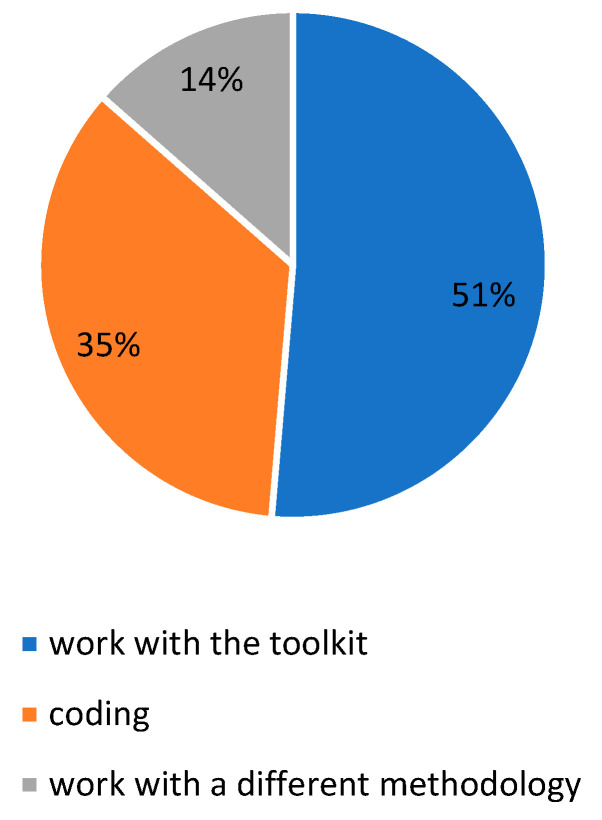
The picture shows the responses to the question “What did you like the most about the experience?”.

**Table 1 biomimetics-08-00161-t001:** Robot geometric and kinematic parameters resulting from the preliminary sizing.

*U* [mm/s]	*f*	θ_0_	*c* [mm]	*S_fin_* [m^2^]	*K_T_*	*C_D_*	*A_f_* [m^2^]
100	1.3	15°	60	0.035	3.6	0.5	0.04

**Table 2 biomimetics-08-00161-t002:** Cruising performance: steady forward velocity at different tail oscillation frequencies.

Tail Frequency [Hz]	Cruising Velocity [mm/s]	Required Torque [Nm]	Available Torque [Nm]
0.75	58	0.017	0.143
1.00	80	0.027	0.104
1.20	92	0.040	0.073
1.30	101	0.047	0.057
1.50	115	0.061	0.026
1.67	130	0.076	-

**Table 3 biomimetics-08-00161-t003:** Maneuvering performance: turnabout radius, in BL, at different speed and steering angle.

Steering Angle [Deg]	Swimming Speed [mm/s]		
	80	90	100
15	1.99	2.53	2.98
30	1.83	2.41	2.80
45	1.74	2.29	2.73

**Table 4 biomimetics-08-00161-t004:** Outline of the activities carried out during the summer camp.

Day	Topics	Tool
1	The robotic disciplines The robotics jobs The Hardware and Software tools for robot design	Basic kit
2	My first machine The robot brain	Basic kit
3	Senses and Sensors Muscles and actuators Environments peculiarities	Basic kit
4	Marine robotics	Fish robot
5	Sustainability	Fish robot
6	Sharing the lesson learned	Basic kit and fish robot

**Table 5 biomimetics-08-00161-t005:** Items of the questionnaire before (BL) and after (PT) the educational activities.

BL	PT	Response Option/s	Dimension
Why did you choose to participate in this experience?	-	Out of curiosity. I will improve my understanding of robotics; I like robotics	Motivation
-	What did you like the most about the activities?	Work with the toolkit; coding/work in a different learning environment	Overall satisfaction
-	How would you rate this experience?	1 (low) to 4 (high)	Overall satisfaction
-	My opinion about robotics changed after carrying out the activities.	1 (disagree) to 4 (strongly agree)	Overall satisfaction
(BL4) The activities will be fun and engaging.	(PT4) The activities were fun and engaging.	1 (low) to 4 (high)	Interest in robotics
(BL6) It will be easy to deal with the robotic toolkit.	(PT5)It was easy to deal with the robotic toolkit.	1 (low) to 4 (high)	Self-efficacy
What would you like to do with the things you will learn at the course?	What would you like to do with the things you learned at the course?	Free text (recoded after the main themes of the responses)	Interest in sustainability

**Table 6 biomimetics-08-00161-t006:** Manufacturing costs per robot unit and training course costs per teacher.

	Costs per Robot Unit per Teacher [EUR]	
	Authors’ Lab	Third-Party Facility
Robot structure	62	30
Mechatronic components	70	-
Mechanical components		-
Robot assembly	20	-
Training course (5 h per 5 days)	50	-
Total	212	180

## Data Availability

Data are not available due to ethical and privacy restrictions.

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
