# Peer review of "Disseminating STEM Subjects and Ocean Literacy through a Bioinspired Toolkit"

_biomimetics, 2023, doi:10.3390/biomimetics8020161_

Round 1

Reviewer 1 Report

Dear Authors,

This is an interesting and very relevant manuscript on the use of a fish-robot as a toolkit for use in pedagogical applications for in schools. I have three minor revisions.

Minor: 1. Please detail in Figures 7 &8 what BL and PT stand for again. It is recommended that BL is before PT in the dataset for ease of visual interpretation of the results and that number of students (n=25) is added to the figure captions.

2. The final sentence of abstract (line 22/23) does not match the conclusions or perhaps more information on the sustainability workshop element is required. In the questionnaire provided it does not seem to assess how the toolkit shifted attention as concluded (line 492). The open text responses of Q7 was an outcome but this was not assessed like the self-efficiency and interest in activities was. As the counter argument that any robot could have potentially provided this shift arises, please elaborate how the water and animal interaction was explored to provide this shift, and the 'vital ideas related to sustainability'. 

3. This is a very useful novel fish-robot for use in STEM teaching and especially as an open education resource, please also include other fish-robot toolkits on the market for exploring STEM in schools as part of your introduction/literature review.

Reviewer 2 Report

The article “Disseminating STEM Subjects and Ocean Literacy Through a Bio-inspired Toolkit” describes a robotic fish system and course curriculum for teaching about robotics, fluid dynamics, and ocean/environmental applications. Overall, the paper provides compelling information on the design of a simple, safe, and easy-to-use kit to teach primary grade school students through teacher training, as well as a case study in one school showing positive interactions among the toolkit with its students. However, there are many questions that the authors have not addressed regarding how the toolkit will be implemented. Therefore, I recommend major revisions before potential publication as an Article in Biomimetics. More detailed comments are listed below:

1.     There is an excellent consideration for the safety features, as described in the methods. The description of the robot, toolkit, and safety considerations are well thought.

2.     What is the current cost per unit of manufacturing the robot, and do you have estimates of how the price per unit would decrease if it were to be scaled up? How much would the costs be to actually implement this, including the amount of time and money to train teachers?

3.     Both the basic and advanced courses sound like a significant amount of time and learning for the school teachers in advance of teaching the modules themselves. For the case study, were real teachers used, or did the authors train the students directly?  If teachers were used, how much training did the teachers require (hours, days, etc.)? I see the time it took to teach the students, but there is no information about how long it took to train the teachers.

4.     In line 273, “dismiss such systems” is a confusing statement. What does this mean?

5.     You mention the different color-coded lessons in the text and show an image with the colors in Figure 5. However, what do the colors actually mean? There should be a legend or key describing this in more detail.

6.     Add a scale bar to Figures 1 and 3.

7.     What do the numbers mean for PT4, BL4, PT5, and BL6 in Figures 7 and 8? The numbers are very confusing. Also, BL and PT are defined previously, but they should be defined again in the figure captions directly to aid the reader (as they are redefined in Figure 9). Similarly, the axes need be labeled, not just described in the text.

8.     The phrase, “As a matter of fact,” is overused throughout the paper.

9.     Moderate English editing is required.

To summarize, this work is important and would add value to the educational system. However, the paper in its current form is missing a discussion about the cost (both time and money) it would take to successfully implement this program. It is understood that any program of this caliber requires substantial time and money to create and implement in its early stages, but this is a limitation that should be explicitly stated in the manuscript. If all of these concerns are sufficiently addressed by the authors, then this paper would be of interest to the readers of Biomimetics and the broader scientific community.

Round 2

Reviewer 2 Report

The authors have addressed my comments sufficiently, and I am happy to recommend publication in Biomimetics.